# Why Matter Matters: Fast-Tracking *Mycobacterium abscessus* Drug Discovery

**DOI:** 10.3390/molecules27206948

**Published:** 2022-10-17

**Authors:** Uday S. Ganapathy, Thomas Dick

**Affiliations:** 1Center for Discovery and Innovation, Hackensack Meridian Health, Nutley, NJ 07110, USA; 2Department of Medical Sciences, Hackensack Meridian School of Medicine, Nutley, NJ 07110, USA; 3Department of Microbiology and Immunology, Georgetown University, Washington, DC 20007, USA

**Keywords:** *Mycobacterium abscessus*, *Mycobacterium tuberculosis*, de novo drug discovery, chemical matter

## Abstract

Unlike Tuberculosis (TB), *Mycobacterium abscessus* lung disease is a highly drug-resistant bacterial infection with no reliable treatment options. De novo *M. abscessus* drug discovery is urgently needed but is hampered by the bacterium’s extreme drug resistance profile, leaving the current drug pipeline underpopulated. One proposed strategy to accelerate de novo *M. abscessus* drug discovery is to prioritize screening of advanced TB-active compounds for anti-*M. abscessus* activity. This approach would take advantage of the greater chance of homologous drug targets between mycobacterial species, increasing hit rates. Furthermore, the screening of compound series with established structure–activity-relationship, pharmacokinetic, and tolerability properties should fast-track the development of in vitro anti-*M. abscessus* hits into lead compounds with in vivo efficacy. In this review, we evaluated the effectiveness of this strategy by examining the literature. We found several examples where the screening of advanced TB chemical matter resulted in the identification of anti-*M. abscessus* compounds with in vivo proof-of-concept, effectively populating the *M. abscessus* drug pipeline with promising new candidates. These reports validate the screening of advanced TB chemical matter as an effective means of fast-tracking *M. abscessus* drug discovery.

## 1. Introduction

When it comes to bacterial infections, *Mycobacterium abscessus* lung disease ranks among the worst. There is no treatment regimen that offers a reliable cure for *M. abscessus* lung disease [1,2], with average cure rates of just 50% [3]. This dire situation stems from the fact that *M. abscessus* displays intrinsic resistance to a wide range of antibiotics, including all first line drugs for Tuberculosis (TB) [4,5,6,7]. The bacterium’s expression of enzymes that either modify antibiotics or their drug targets contributes significantly to its extreme drug resistance profile [5,8,9,10,11,12,13]. As a result, treating *M. abscessus* lung disease is as difficult as extensively drug-resistant TB. Current treatments typically combine an oral macrolide (clarithromycin or azithromycin) with parenteral drugs, including amikacin with either tigecycline, imipenem, or cefoxitin [1,2,14]. With treatment requiring 18–24 months to complete, serious drug side effects are often encountered, contributing to poor compliance and the emergence of acquired drug resistance. Treatment is further hampered by the presence of *erm41* inducible macrolide resistance in this bacterium [15] and the need for intravenous drug administration.

Given the limitations of current treatment, new drugs and treatment options are urgently needed for *M. abscessus* lung disease [16]. Progress has been made on drug reformulation to improve the current treatment regimen [17,18]. Drug repurposing has also been proposed to quickly bolster treatment options for patients [19,20,21]. Drug repositioning could restore functionality to drug classes inactivated by *M. abscessus* intrinsic drug resistance mechanisms [22,23]. In addition to these efforts, de novo *M. abscessus* drug discovery should be pursued. New chemical matter with either novel targets or mechanisms of action could address intrinsic and acquired drug resistance. Furthermore, novel antibiotics with optimized anti-*M. abscessus* activity could shorten treatment times, reducing harmful side effects and the development of drug resistance.

Unfortunately, the *M. abscessus* drug pipeline remains underpopulated [24,25,26]. The situation is perhaps unsurprising given the inherent challenges of de novo drug discovery (Figure 1). The process is inefficient, requiring screens of large compound libraries to compensate for low hit rates. While screens may identify compounds with whole cell activity, these hits may show no structure–activity relationship (SAR) and/or have poor pharmacokinetic properties (PK). These pitfalls contribute to high rates of attrition in the hit-to-lead optimization phase. As a result, de novo drug discovery requires massive investments in time and resources. In the case of *M. abscessus*, the efficiency of de novo drug discovery may be further reduced by the bacterium’s high intrinsic drug resistance. Indeed, a recent screen of a diverse chemical library had a ten-fold lower hit rate against *M. abscessus* compared to the more drug-susceptible *M. tuberculosis* [27]. Instead of engaging in conventional high-throughput whole cell screening, de novo *M. abscessus* drug discovery will require a different approach.

Recently, we proposed a two-part strategy to accelerate de novo *M. abscessus* drug discovery. First, we selectively screen TB active compounds for anti-*M. abscessus* activity. As a member of the non-tuberculous mycobacteria (NTM), *M. abscessus* is a close relative of *M. tuberculosis* [28]. Therefore, TB actives are likely to penetrate the mycobacterial cell envelope and have a homologous, structurally conserved target in *M. abscessus*, which would increase hit rates. In fact, a screen of TB active compounds did yield a higher hit rate when compared to a random compound library [29]. Second, we focus screening on advanced compounds. With established SAR, PK, and tolerability, advanced compounds can readily move from in vitro anti-*M. abscessus* hits to lead compounds with in vivo efficacy, lowering early attrition rates. Given the progress made over the past 20 years, TB drug discovery has produced numerous advanced compound series, many of which are still in active development. Therefore, the screening of advanced TB chemical matter has untapped potential to fast track de novo *M. abscessus* drug discovery by populating the *M. abscessus* drug pipeline with advanced lead compounds (Figure 1). To evaluate this hypothesis, we examined the literature. We found several reports in which the screening of focused collections of advanced TB actives led to the identification of anti-*M. abscessus* advanced lead compounds (Table 1). As our focus is on the fast-tracking of novel preclinical candidates for *M. abscessus* lung disease, we highlight in this review those compounds for which in vivo proof-of-concept was demonstrated in an animal lung infection model. While it should be noted that some TB active compounds have demonstrated efficacy against *M. abscessus* in other in vivo models [30], these examples will not be discussed here.

## 2. EC/11770 (Benzoxaboroles)

Benzoxaboroles are a class of boron-heterocyclic molecules that target the editing domain of leucyl-tRNA synthetase (LeuRS) via the oxaborole tRNA-trapping (OBORT) mechanism, inhibiting protein synthesis [39,40]. After antifungal activity was reported for tavaborole [39], benzoxaboroles were optimized for antibacterial activity. The addition of a 3-aminomethyl group and a 7-*O*-propanol to the benzoxaborole core resulted in epetraborole, which has broad activity against Gram-negatives [41,42]. The subsequent addition of a 4-halogen group (either chlorine or bromine) led to enhanced anti-TB activity [43]. *M. tuberculosis* mutants selected for resistance to this series of benzoxaboroles had mutations in *leuS*, suggesting that the target was the mycobacterial LeuRS [43]. Further SAR activities enabled the development of GSK3036656 (GSK656), which had improved in vivo PK and was active in a mouse model of TB infection [44]. Given these properties, EC/11770, a close analog of GSK656, was tested for anti-*M. abscessus* activity (Table 1) [31]. EC/11770 was active against all three subspecies of the *M. abscessus* complex (subsp. *abscessus*, subsp. *bolletii*, and subsp. *massiliense*) and retained activity against *M. abscessus* biofilms [31]. Since benzoxaboroles had been previously characterized as LeuRS inhibitors in fungi [40] and *M. tuberculosis* [43], *M. abscessus* target deconvolution for EC/11770 proceeded more quickly [31]. After selecting for *M. abscessus* EC/11770-resistant mutants on agar, targeted sequencing of *leuS*, the gene encoding the *M. abscessus* LeuRS homolog, was performed. As observed in other species [40,43], all *M. abscessus* EC/11770-resistant mutants carried point mutations in the editing domain of LeuRS [31], suggesting that EC/11770 targets this enzyme via the previously reported OBORT mechanism [40]. As an advanced compound, EC/11770 also displayed attractive PK properties including high oral bioavailability that enabled further characterization in vivo. Indeed, EC/11770 was efficacious against *M. abscessus* in a mouse lung infection model, with a low 10 mg/kg dose being sufficient to match a 250 mg/kg dose of the standard of care macrolide clarithromycin [31]. Thus, the screening of TB active benzoxaboroles yielded EC/11770 as a preclinical candidate for *M. abscessus* lung disease. This effort further benefitted *M. abscessus* drug discovery by identifying a new drug class (benzoxaboroles) and drug target (LeuRS).

The non-halogenated, anti-Gram-negative epetraborole was also shown to have in vitro activity against the *M. abscessus* complex that was comparable to EC/11770 [45]. This benzoxaborole was also active in murine *M. abscessus* infection models in two independent studies [45,46]. Similarly, MRX-6038, a member of a novel series tricyclic benzoxaboroles, was reported to be active against *M. abscessus* complex in vitro as well as in mouse lungs [47]. These findings expand the potential for a benzoxaborole-based treatment for *M. abscessus* lung disease.

## 3. IC25 (Indole 2-Carboxamides)

Indole 2-carboxamides (ICs) were identified as TB actives in multiple screens [48,49,50]. These compounds target the essential *M. tuberculosis* lipid transporter MmpL3, disrupting the translocation of trehalose mono-mycolate (TMM) across the inner membrane and, subsequently, cell wall biosynthesis [51,52,53,54]. The 4,6-dimethyl indole IC25 (Table 1) was characterized as a promising advanced compound with nanomolar anti-TB potency, low cytotoxicity, good oral bioavailability, and in vivo efficacy [51,55]. IC25 was then selected for testing against *M. abscessus*. IC25 retained potent activity against the *M. abscessus* complex in vitro and in macrophages [56,57]. As observed in *M. tuberculosis*, IC25 inhibited the transport of mycolic acids in *M. abscessus* [57]. IC25 was also 16 times less potent against an isogenic *M. abscessus* strain carrying a missense mutation in MmpL3 [32]. This combination of biochemical and genetic evidence suggests that IC25 retains the MmpL3 transporter of *M. abscessus* as its target [32]. IC25 was also efficacious in a murine model of *M. abscessus* lung infection, with a 300 mg/kg dose being as effective as a 150 mg/kg dose of amikacin [32]. These findings support IC25 as a preclinical candidate for *M. abscessus* lung disease and validate MmpL3 as an *M. abscessus* drug target.

An unsubstituted indole, IC5, was also active against *M. abscessus* in vivo [32]. Therefore, a greater range of IC derivatives may have clinical potential. Indeed, recent SAR activities have generated many IC analogs with not just anti-*M. abscessus* activity but desirable PK properties and intramacrophage activity [56,57], further increasing the chances of an IC-based treatment option for *M. abscessus* lung disease.

## 4. Cyclohexyl-Griselimycin (Cyclic Depsipeptides)

The cyclic depsipeptide griselimycin is a natural product extracted from *Streptomyces* [58,59,60]. When griselimycin was found to have anti-tubercular activity in the 1960s, it was pursued as a candidate TB drug. Unfortunately, its development was stopped due to poor oral bioavailability in initial human clinical trials [61]. Since a griselimycin derivative was active against drug-resistant TB [62], an optimization campaign was recently pursued for griselimycin to improve its potency and pharmacological properties [63]. Griselimycin’s Pro^8^ residue was determined to be the site of its metabolic degradation, and modifications at this site not only provided greater metabolic stability and in vivo exposure but enhanced anti-TB potency in vitro [63]. Compared to griselimycin, the cyclohexyl derivative cyclohexyl-griselimycin (CGM) (Table 1) was 17 times more potent against *M. tuberculosis* and had bactericidal activity against this pathogen [63]. CGM also displayed optimized PK including higher oral bioavailability, reduced clearance, and a longer half-life [63]. Interestingly, the frequency of resistance (FoR) in mycobacteria to CGM was extremely low (10^−10^ per CFU) and was linked to a novel mechanism of action [63]. CGM can bind to a peptide interaction site of the DNA sliding clamp DnaN, disrupting protein–protein interactions with the polymerase III α subunit DnaE1 and proteins involved in DNA repair [63,64,65]. Crucially, CGM demonstrated in vivo activity in murine TB infection models when administered alone or in combination with other TB drugs [63]. Thus, CGM is not only promising as a potential TB drug but represents a prime candidate to screen for anti-*M. abscessus* activity.

Recently, CGM was found to be active against a panel of reference strains and clinical isolates from the *M. abscessus* complex [33]. As observed against *M. tuberculosis*, CGM retained bactericidal activity against *M. abscessus* [33]. At the same 250 mg/kg dosage, CGM was as effective as clarithromycin against *M. abscessus* in a mouse model [33]. Thus, CGM has newfound potential as a preclinical candidate for *M. abscessus* lung disease. These findings also establish DnaN as a novel drug target for *M. abscessus* drug development.

## 5. OZ439 (Synthetic Trioxolanes)

When faced with hypoxic conditions, mycobacteria can enter a dormant, non-replicating state that maintains their viability and confers tolerance to many antibiotics [66]. This dormancy response is associated with expression of a 50+ gene dormancy regulon that is controlled by the transcriptional regulator DosR and two sensor histidine kinases, DosS and DosT [67]. The antimalarial drug artemisinin was identified in a screen for inhibitors of the *M. tuberculosis* DosRST pathway [68]. By targeting the heme associated with DosS and DosT, artemisinin impairs the ability of *M. tuberculosis* to survive under hypoxic conditions and tolerate antibiotics [68].

Synthetic trioxolanes are a new generation of antimalarials that are based on artemisinin but feature optimized PK and safety profiles [69]. Given the TB activity of artemisinin, synthetic trioxolane OZ439 was recently evaluated against *M. abscessus* (Table 1) [34]. OZ439 inhibited DosRS signaling in *M. abscessus* and reduced the heme of *M. abscessus* DosS [34]. OZ439 treatment also reduced *M. abscessus* drug tolerance under hypoxia and inhibited *M. abscessus* biofilm formation. Critically, OZ439 was active in both acute and chronic *M. abscessus* murine infection models [34]. Similar in vivo efficacy was obtained with a close analog, OZ277 [34]. Interestingly, OZ439 also potentiated the activity of two *M. abscessus* standard-of-care drugs (amikacin and azithromycin) in vivo [34], demonstrating compatibility of synthetic trioxolanes with *M. abscessus* combination therapy.

## 6. EC/11716 (Novel Bacterial Topoisomerase Inhibitors)

The mycobacterial DNA gyrase is a type IIA DNA topoisomerase and a validated anti-mycobacterial drug target. Fluoroquinolones, a class of gyrase inhibitors, are used to treat drug-resistant TB infections [70]. These compounds target the cleavage-ligation active site of DNA gyrase, creating stalled enzyme-DNA cleavage complexes. This action generates double-stranded DNA breaks that kill the bacterium [71]. Unfortunately, *M. abscessus* displays intrinsic resistance to fluoroquinolones, preventing these compounds from being effectively used to treat *M. abscessus* lung infections [72,73,74]. Thus, the identification and development of alternative gyrase inhibitors with anti-*M. abscessus* activity is being actively pursued [75,76].

Novel bacterial topoisomerase inhibitors (NBTIs) are a new class of gyrase inhibitors with activity against both Gram-positive and Gram-negative bacteria [77,78,79,80,81,82,83,84]. These compounds feature left-hand side (LHS) and right-hand side (RHS) portions that are connected by a central unit (CU). These structural features allow NBTIs to target DNA gyrase via a different mechanism of action from that of fluoroquinolones. The RHS binds to a transient pocket between the two GyrA subunits while the LHS intercalates into the DNA midway between the two sites of DNA cleavage [81,85]. This action stabilizes enzyme-DNA cleavage complexes that generate single-stranded DNA breaks instead of double-stranded ones [85,86].

*Mycobacterium tuberculosis* gyrase inhibitors (MGIs) are a subset of NBTIs with anti-TB activity [87]. These compounds adapt the NBTI scaffold with a 7-substituted-1,5-naphthyridin-2-one in the LHS, an aminopiperidine as the CU, and a monocyclic aromatic ring in the RHS. MGI-resistance in *M. tuberculosis* was associated with GyrA and GyrB mutations, establishing the mycobacterial DNA Gyrase as the drug target [87]. Like other NBTIs, MGIs induced single-stranded DNA breaks in *M. tuberculosis* and were bactericidal against this species [86,87]. These compounds were also active against fluoroquinolone-resistant *M. tuberculosis* strains and in a murine TB infection model [87].

EC/11716, an advanced MGI with TB activity, was recently tested for activity against *M. abscessus* (Table 1) [35]. EC/11716 showed in vitro activity against all subspecies of the *M. abscessus* complex [35]. This MGI retained bactericidal activity against *M. abscessus* and was active against drug-tolerant biofilms [35]. Furthermore, the selection of EC/11716-resistant mutants confirmed DNA gyrase as the drug target and revealed a low FoR [35]. EC/11716 was also active against *M. abscessus* in mice, providing in vivo proof-of-concept [35]. Thus, MGIs like EC/11716 represent alternative gyrase inhibitor-based drug candidates for *M. abscessus* lung disease that would overcome the present limitations of fluoroquinolones.

## 7. TPP8 (Tricyclic Pyrrolopyrimidines)

Developed using a structure-based drug design approach, tricyclic pyrrolopyrimidines (TPPs) are broad-spectrum antibacterial agents that target the ATPase domain of DNA gyrase subunit B [88]. Consistent with having a different mechanism of action from fluoroquinolones, TPPs retained activity against fluoroquinolone-resistant strains of several Gram-positive and Gram-negative pathogens [88]. Further optimization produced analogs that inhibited the *M. tuberculosis* DNA gyrase and were active against drug-resistant TB strains [89,90]. From this series, TPP8 was evaluated for anti-*M. abscessus* activity (Table 1) [36]. TPP8 had potent activity against *M. abscessus* reference strains and clinical isolates representing all three subspecies [36]. In fact, this compound was more potent against *M. abscessus* than two other gyrase inhibitors, moxifloxacin and SPR719 [36]. TPP8 inhibited the supercoiling activity of *M. abscessus* DNA gyrase in vitro, and in vitro generated *M. abscessus* TPP8-resistant strains had mutations in the ATPase domain of GyrB, confirming the drug’s target and mechanism of action [36]. TPP8 also inhibited the intracellular growth of *M. abscessus* in macrophages [36]. Due to TPP8’s lack of oral bioavailability, intraperitoneal administration was used to evaluate the compound’s in vivo efficacy in NOD SCID mice [36]. A 25 mg/kg dose of TPP8 reduced the lung bacterial burden by 20-fold and was as effective as treatment with clarithromycin (at 250 mg/kg) or moxifloxacin (at 200 mg/kg), providing in vivo proof-of-concept for TPPs as anti-*M. abscessus* agents [36].

## 8. CRS400226 (Benzothiazole Amides)

Benzothiazole adamantyl amide was identified as an anti-*M. abscessus* hit in a screen of a focused library of TB active compounds [91]. This compound shares structural features with other MmpL3 inhibitors (e.g., indole 2-carboxamides) suggesting a common mechanism of action. Indeed, benzothiazole amide analogs alter mycolic acid transport in *M. abscessus*, and spontaneous *M. abscessus* mutants resistant to these compounds had nonsynonymous mutations in *mmpl3* [37]. SAR studies produced the optimized analog CRS400226 (Table 1), which had improved in vitro potency against *M. abscessus* and was bactericidal against this pathogen [37,92]. In addition to having good metabolic stability, CRS400226 displayed low levels of cytotoxicity, cytochrome P450 inhibition, and hemolysis [37]. Due to the inherent hydrophobicity of benzothiazole amides, CRS400226 and other lead analogs showed low solubility, high protein binding, and lacked oral bioavailability, preventing them from becoming oral drug candidates [37]. Nonetheless, intratracheal administration of CRS400226 achieved efficacy in an *M. abscessus* mouse infection model that was on par with azithromycin treatment [37].

## 9. TBAJ-876 (Diarylquinolines)

Bedaquiline was the first diarylquinoline reported with potent activity against drug-sensitive and drug-resistant *M. tuberculosis* [93]. By targeting the proton pump of the mycobacterial F-ATP synthase (AtpE), this compound blocks the bacterium’s ability to produce ATP, killing the bacterium both in vitro and in mouse models [93,94]. As the first new TB drug approved in four decades [95], bedaquiline is currently used to treat multidrug-resistant TB [96] and as salvage therapy for *M. abscessus* lung disease [97]. This drug, however, has known PK and toxicity issues. Pharmacologically, bedaquiline displays a long half-life, tissue accumulation, and high lipophilicity (cLogP = 7.25) [98,99]. The compound also inhibits the cardiac potassium channel protein hERG, resulting in prolongation of the QT interval and increased risk of death in patients with cardiac ischemia [98].

An optimization campaign to address bedaquiline’s liabilities found that replacing its C-unit naphthalene with a 3,5-dialkoxy-4-pyridyl group significantly reduced hERG inhibition [100]. These 3,5-dialkoxypyridine analogs of bedaquiline also displayed lower lipophilicity and improved anti-TB potency [100]. From this series, TBAJ-876 was selected for further development (Table 1). Compared to bedaquiline, TBAJ-876 displays lower hERG channel inhibition, higher clearance, and greater in vitro potency against *M. tuberculosis* [100]. TBAJ-876 also targets F-ATP synthase via the same mechanism of action as bedaquiline and is similarly bactericidal against *M. tuberculosis* [101,102]. Furthermore, TBAJ-876 showed efficacy in a mouse model of TB infection [100], supporting this compound as a preclinical development candidate for TB.

Recently, TBAJ-867 was evaluated for anti-*M. abscessus* activity [38]. TBAJ-867 had potent activity against reference strains and clinical isolates representing all three subspecies of the *M. abscessus* complex [38]. Notably, TBAJ-876 did not antagonize the activity of several commonly used anti-*M. abscessus* drugs in checkerboard assays, suggesting potential for co-administration in future treatment regimens [38]. Unlike with TB, TBAJ-876 was bacteriostatic against *M. abscessus* [38], but similar results were observed with bedaquiline against this NTM species [103]. Nonetheless, TBAJ-876 was active in a murine model of *M. abscessus* infection, with a 10 mg/kg dose being as effective as administering bedaquiline [38]. Thus, the TB drug candidate TBAJ-876 has expanded potential as a treatment for *M. abscessus* lung disease.

## 10. Conclusions

While de novo drug discovery is urgently needed for *M. abscessus* lung disease, these efforts remain limited. One strategy to enhance *M. abscessus* drug discovery proposes focused screening of advanced TB chemical matter. As demonstrated in this review, recent screens of TB active compounds for anti-*M. abscessus* activity have yielded no less than eight new compounds as advanced leads for *M. abscessus* lung disease with in vivo proof-of-concept in mouse models of infection (Table 1). These compounds come from a diverse set of drug classes and are associated with a broad range of targets. In fact, several of these compounds provide the first pharmacological validation of new *M. abscessus* drug targets (e.g., EC/11770, OZ439, and cyclohexyl-griselimycin), providing new ways to overcome *M. abscessus*’s drug resistance and new opportunities for target-based drug discovery. Other compounds target previously known drug targets but offer other advantages. For instance, EC/11716 and TPP8 target DNA gyrase via novel mechanisms of action that would address *M. abscessus*’s intrinsic resistance to fluoroquinolones. The ATP synthase inhibitor TBAJ-876 also improves upon bedaquiline with better PK and reduced toxicity. Such desirable properties make the newly identified compounds attractive candidates for further development.

Altogether, recent screens of TB chemical matter have greatly helped to populate the *M. abscessus* preclinical drug pipeline, validating this strategy as an efficient approach to de novo drug discovery for *M. abscessus* lung disease. With numerous compound libraries generated over the past two decades of TB drug discovery, we expect further screening of advanced TB actives to sustain the *M. abscessus* drug pipeline for years to come.

## Figures and Tables

**Figure 1 molecules-27-06948-f001:**
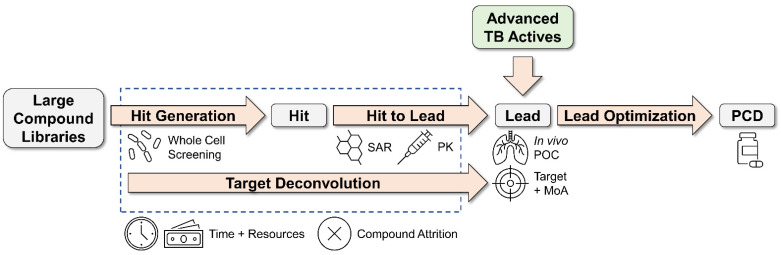
Fast tracking of de novo *M. abscessus* drug discovery by screening advanced TB chemical matter. Conventional de novo drug discovery requires the screening of large compound libraries to generate hits with whole cell activity. For *M. abscessus*, which displays high intrinsic drug resistance, hit rates from whole cell screens are lower than those observed for *M. tuberculosis* [27]. Subsequent hit to lead optimization involves structure–activity relationship (SAR) studies and preliminary pharmacokinetic profiling (PK) to identify lead compounds with improved potency and, ultimately, in vivo proof-of-concept (POC) in animal lung infection models. Target deconvolution is performed in parallel to identify the drug target and mechanism of action (MoA). This involves the selection of *M. abscessus* drug-resistant mutants, target identification by whole genome sequencing, and target validation by biochemical assays, genetics, and structural studies. Together, the early phase of de novo drug discovery (inside dashed blue box) is not only time and resource-intensive but subject to compound attrition, as initial hits may lack SAR or have poor PK. By screening libraries of advanced TB active compounds instead, the time required to go from initial screen to lead compound is greatly shortened, providing a fast track to candidates for lead optimization and preclinical development (PCD).

**Table 1 molecules-27-06948-t001:** TB active compounds identified as advanced lead compounds for *M. abscessus* lung disease.

Compound	Structure	Class	Target	Pathway	Reference *^a^*
EC/11770	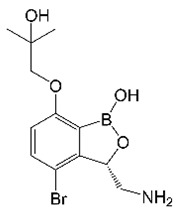	Benzoxaborole	Leucyl-tRNA synthetase	Protein Biosynthesis	[31]
IC25	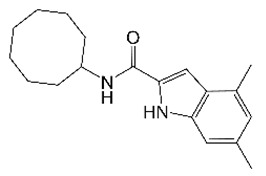	Indole 2-carboxamide	Membrane Transporter MmpL3	Cell Wall Biosynthesis	[32]
Cyclohexyl-griselimycin	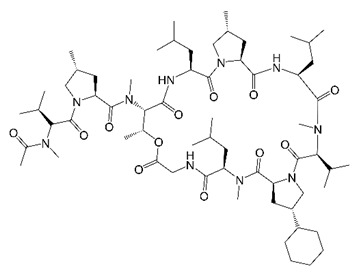	Cyclic depsipeptide	DNA Sliding Clamp DnaN	DNA Replication	[33]
OZ439	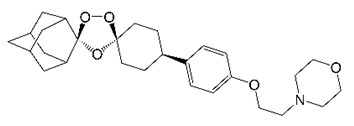	Synthetic trioxolane	Sensor Histidine Kinase DosS	Dormancy Response	[34]
EC/11716	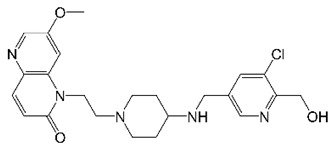	NBTI	DNA Gyrase	DNA Topology	[35]
TPP8	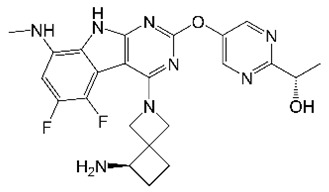	Tricyclic pyrrolo- pyrimidine	DNA Gyrase	DNA Topology	[36]
CRS400226	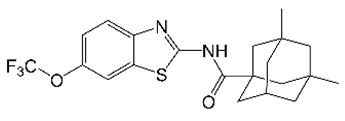	Benzothiazole amide	Membrane Transporter MmpL3	Cell Wall Biosynthesis	[37]
TBAJ-876	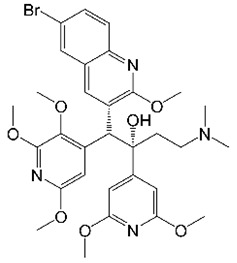	Diarylquinoline	ATP Synthase	ATP Synthesis	[38]

*^a^* Publication in which in vivo proof-of-concept was demonstrated. See text for further information.

## Data Availability

All information was available in the public domain.

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
