# Peer review of "Why Matter Matters: Fast-Tracking Mycobacterium abscessus Drug Discovery"

_molecules, 2022, doi:10.3390/molecules27206948_

Round 1

Reviewer 1 Report

The authors wrote and captured all the biological results in a well-organized way, which can help readers to understand easily.

I suggest to the authors add figure on the first page, helping the readers to understand the background of this work.

In table 1 compound structure bonds vary from one to another one. Please make all structure should be in one format. 

Reviewer 2 Report

The manuscript titled “Why Matter Matters: Fast-Tracking Mycobacterium abscessus Drug Discovery” reports an interesting work and authors focusing on the fast-tracking of novel preclinical candidates for Mycobacterium abscessus lung disease, they highlight the compounds for which in vivo proof-of-concept was demonstrated in an animal lung infection model. I include my comments below, most of them are suggestions to improve the overall quality of the publication. I considered the manuscript suitable for publication subject to the following improvements.

Overall, the study is well presented in a good way. However, some of the sentences include repetitive words and are not explained and cited appropriately.

Add a schematic diagram that may reveal both your proposed strategies and limitations of current treatments and drug discovery procedures.

The authors mentioned different drugs/molecules against M. abscessus, whereas, they didn’t mention the source or origin of the molecules.

Add future prospects and how computational methods can be used in such discoveries.
